# SKIP-CONNECTION AND BATCH-NORMALIZATION IMPROVE DATA SEPARATION ABILITY

## ABSTRACT

The ResNet and the batch-normalization (BN) achieved high performance even when only a few labeled data are available. However, the reasons for its high performance are unclear. To clear the reasons, we analyzed the effect of the skip-connection in ResNet and the BN on the data separation ability, which is an important ability for the classification problem. Our results show that, in the multilayer perceptron with randomly initialized weights, the angle between two input vectors converges to zero in an exponential order of its depth, that the skip-connection makes this exponential decrease into a sub-exponential decrease, and that the BN relaxes this sub-exponential decrease into a reciprocal decrease. Moreover, our analysis shows that the preservation of the angle at initialization encourages trained neural networks to separate points from different classes. These imply that the skip-connection and the BN improve the data separation ability and achieve high performance even when only a few labeled data are available.

## 1 INTRODUCTION

The architecture of a neural network heavily affects its performance especially when only a few labeled data are available. The most famous example of one such architecture is the convolutional neural network (CNN) (LeCun et al., 1995). Even when convolutional layers of CNN were randomly initialized and kept fixed and only the last fully-connected layer was trained, it achieved a competitive performance compared with the traditional CNN (Jarrett et al., 2009; Zhang & Suganthan, 2017). Recent other examples are the ResNet (He et al., 2016) and the batch-normalization (BN) (Ioffe & Szegedy, 2015). The ResNet and the BN are widely used in few-shot learning problems and achieved high performance (Munkhdalai et al., 2018; Oreshkin et al., 2018).

One reason for the success of neural networks is that their architectures enable its feature vector to capture prior knowledge about the problem. The convolutional layer of CNN enable its feature vector to capture statistical properties of data such as the shift invariance and the compositionality through local features, which present in images (Zeiler & Fergus, 2014). However, effects of the skip-connection in ResNet and the BN on its feature vector are still unclear.

To clear the effects of the skip-connection and the BN, we analyzed the transformations of input vectors by the multilayer perceptron, the ResNet, and the ResNet with BN. Our results show that the skip-connection and the BN preserve the angle between input vectors. This preservation of the angle is a desirable ability for the classification problem because the last output layer should separate points from different classes and input vectors in different classes have a large angle (Yamaguchi et al., 1998; Wolf & Shashua, 2003). Moreover, our analysis shows that the preservation of the angle at initialization encourages trained neural networks to separate points from different classes. These imply that the skip-connection and the BN improve the data separation ability and achieve high performance even when only a few labeled data are available.

## 2 PRELIMINARIES

### 2.1 NEURAL NETWORKS

We consider the following $L$ layers neural networks, which transform an input vector $x \in \mathbb{R}^D$ into a new feature vector $h^L \in \mathbb{R}^D$ through layers. Let $h^0 = x$ and $\phi(\cdot) = \max(0, \cdot)$ be the ReLU activation function.

**Multilayer perceptron (MLP):**

$$h_i^{l+1} = \phi\left(u_i^{l+1}\right), \qquad u_i^{l+1} = \sum_{j=1}^{D} W_{i,j}^l h_j^l. \tag{1}$$

**ResNet** (Yang & Schoenholz, 2017; Hardt & Ma, 2017) **:**

$$h_i^{l+1} = \sum_{j=1}^{D} W_{i,j}^{l,2} \phi(u_j^{l+1}) + h_i^l, \qquad u_i^{l+1} = \sum_{j=1}^{D} W_{i,j}^{l,1} h_j^l. \tag{2}$$

**ResNet with batch-normalization (BN):**

$$h_i^{l+1} = \sum_{j=1}^{D} W_{i,j}^{l,2} \phi\left(\mathrm{BN}(u_j^{l+1})\right) + h_i^l,$$

$$\mathrm{BN}(u_i^{l+1}) = \frac{u_i^{l+1} - \mathbb{E}\left[u_i^{l+1}\right]}{\sqrt{\mathrm{Var}\left(u_i^{l+1}\right)}}, \qquad u_i^{l+1} = \sum_{j=1}^{D} W_{i,j}^{l,1} h_j^l, \tag{3}$$

where the expectation is taken under the distribution of input vectors in the mini-batch of the stochastic gradient descent (SGD). Without loss of generality, we assume that the variance of input vectors in the mini-batch is one, $\mathrm{Var}\left(x_d\right) = 1$ for all $d \in [D]$.

We analyzed the average behavior of these neural networks when the weights were randomly initialized as follows. In the MLP, the weights were initialized by the He initialization (He et al., 2015) because the activation function is the ReLU function.

$$W_{i,j}^l \sim \mathcal{N}\left(0, \frac{2}{D}\right). \tag{4}$$

In the ResNet and the ResNet with BN, the first internal weights were initialized by the He initialization, but the second internal weights were initialized by the Xavier initialization (Glorot & Bengio, 2010) because the second internal activation function is the identity.

$$W_{i,j}^{l,1} \sim \mathcal{N}\left(0, \frac{2}{D}\right), \qquad W_{i,j}^{l,2} \sim \mathcal{N}\left(0, \frac{1}{D}\right). \tag{5}$$

### 2.2 TRANSFORMATION OF INPUT VECTORS

We analyzed the transformation of input vectors through hidden layers of the neural networks. Now we define the quantity studied in this paper.

**Definition 1** *For a pair of the feature vectors $h^l(n), h^l(m)$ of input vectors $x(n), x(m)$, we define the angle and the cosine similarity,*

$$\angle(h^l(n), h^l(m)) = \arccos\left(c^l(n, m)\right), \qquad c^l(n, m) = \frac{q^l(n, m)}{\sqrt{q^l(n)q^l(m)}} \tag{6}$$

*where $q^l(n) = \mathbb{E}\left[\|h^l(n)\|^2\right]$ is the length of the feature vector and $q^l(n, m) = \mathbb{E}\left[h^l(n)^T h^l(m)\right]$ is the inner product between the pair of the feature vectors. Note that the expectation is taken under the probability distribution of initial weights.*

## 3 MAIN RESULTS

### 3.1 RECURRENCE RELATION OF THE ANGLE

We derived the recurrence relation of the angle (Table 1). Its plot (Fig.1) shows that the MLP contracts the angle between input vectors, which is an undesirable property for the classification

Table 1: Recurrence relation of the angle. Let $\psi(\theta) = \frac{1}{\pi}\left(\sin\theta + (\pi - \theta)\cos\theta\right)$.

| Model | Angle $\angle(h^{l+1}(n), h^{l+1}(m))$ | Cosine similarity $c^{l+1}(n, m)$ |
|---|---|---|
| MLP | | $\psi(\angle(h^l(n), h^l(m)))$ |
| ResNet | $\arccos\left(c^{l+1}(n, m)\right)$ | $\frac{1}{2}\psi(\angle(h^l(n), h^l(m))) + \frac{1}{2}\cos\angle(h^l(n), h^l(m))$ |
| ResNet with BN | | $\frac{1}{l+3}\psi(\angle(h^l(n), h^l(m))) + \frac{l+2}{l+3}\cos\angle(h^l(n), h^l(m))$ |

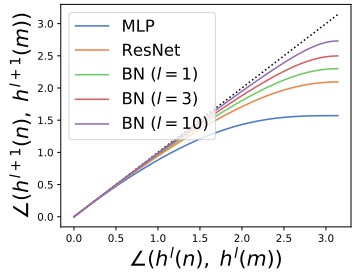
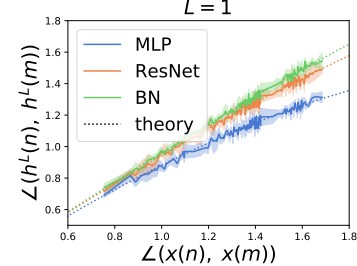
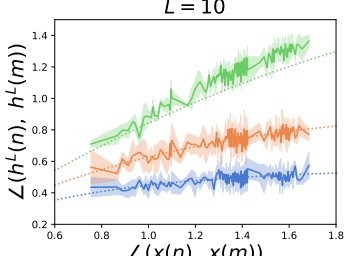

Figure 1: Recurrence relation of the angle. The skip-connection and the BN preserve the angle.

Figure 2: Transformation of the angle between input vectors. We plotted the mean and the standard deviation of the angle over 10 randomly initialized parameters.

problem, and that the skip-connection in ResNet and the BN relax this contraction. Numerical simulations (Fig.2) on the MNIST dataset (LeCun et al., 1998) validated our analysis.

Table 1 gives us the clear interpretation how the skip-connection in ResNet and the BN preserve the angle between input vectors. The ReLU activation function contracts the angle because the ReLU activation function truncates negative value of its input. The skip-connection bypasses the ReLU activation function and thus reduces the effect of the ReLU activation function to the half. Moreover, the BN reduces the effect of the ReLU activation function to the reciprocal of the depth.

### 3.2 DYNAMICS OF THE ANGLE THROUGH LAYERS

We derived the dynamics of the angle through layers (Table 2) by applying the recurrence relation of the angle (Table 1) iteratively and using the fact that, if $\theta$ is small, $\arccos(\psi(\theta))$ can be well approximated by the linear function, $a \cdot \theta$ where $a < 1$ is constant. Table 2 shows that, in the MLP with randomly initialized weights, the angle between input vectors converges to zero in an exponential order of its depth, that the skip-connection in ResNet makes this exponential decrease into a sub-exponential decrease, and that the BN relaxes this sub-exponential decrease into a reciprocal decrease. In other words, the skip-connection in ResNet and the BN preserve the angle between input vectors. Numerical simulation (Fig.3) on the MNIST dataset validated our analysis.

Table 2: Dynamics of the angle through layers.

| Model | Angle $\angle(h^L(n), h^L(m))$ |
|---|---|
| MLP | $\simeq a^L \cdot \angle(x(n), x(m))$ |
| ResNet | $\geq \left(\frac{1+a}{2}\right)^L \cdot \angle(x(n), x(m))$ |
| ResNet with BN | $\geq \frac{2}{L+2} \cdot \angle(x(n), x(m))$ |

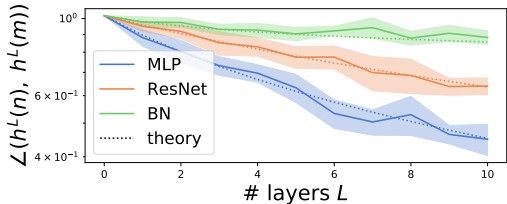

Figure 3: Dynamics of the angle. We plotted the mean and the standard deviation of the angle over 10 randomly initialized parameters.

### 3.3 ROLE OF TRAINING AND ITS RELATION TO THE PRESERVATION OF THE ANGLE

A desirable ability of the neural network for the classification problem is to separate points from different classes. However, our results show that the randomly initialized neural networks contract the angle between input vectors from different classes. Our analysis provide us with an insight how training tackle this problem. We can show that the cosine similarity $c^{l+1}(n, m)$ is proportional to

$$\cos \theta \cos(\theta - \angle(h^l(n), h^l(m))) \tag{7}$$

where $\theta$ is a parameter we can control by training. Its plot (Fig.4) implies that training makes small angles smaller and large angles larger by taking the extreme value of $\theta$ like 0 or $\pi$. In order to validate this insight, we stacked the softmax layer on top of an 1 layer MLP and trained this model by the SGD with 100 labeled examples in the MNIST dataset. Fig.5 shows the change of the angles of feature vectors by training, which validated our insight.

The above discussion also shows the relationship between training and the preservation of the angle. The angle of feature vectors at high layer of the initialized MLP is small, which implies that training doesn't take extreme value of $\theta$ and doesn't separate points from different classes. On the other hand, the skip-connection and the BN preserve the angle between input vectors even at high layer. Thus, training takes extreme value of $\theta$ and separates points from different classes. Numerical simulations (Fig.6), which is the same as the previous one, validated our insight.

## 4 CONCLUSION

The ResNet and the BN achieved high performance even when only a few labeled data are available. To clear the reasons for its high performance, we analyzed effects of the skip-connection in ResNet and the BN on the transformation of input vectors through layers. Our results show that the skip-connection and the BN preserve the angle between input vectors, which is a desirable ability for the classification problem. Moreover, our analysis shows that the preservation of the angle at initialization encourages trained neural networks to separate points from different classes. These results imply that the skip-connection and the BN improve the data separation ability and achieve high performance even when only a few labeled data are available.

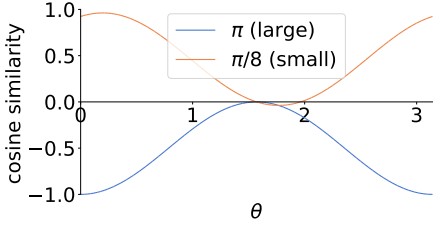

Figure 4: Cosine similarity of feature vectors $c^{l+1}(n, m)$ for vectors $h^l(n), h^l(m)$ which have a large angle or small angle.

Figure 5: Change of the angles of feature vectors by training.

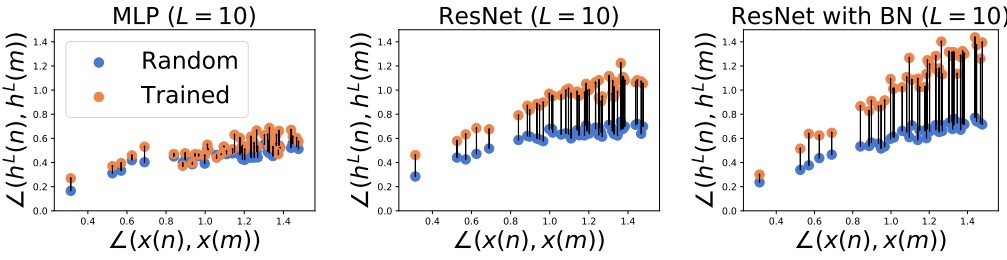

Figure 6: Change of the angles of feature vectors by training

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
