# OpenReview forum: "Skip-connection and batch-normalization improve data separation ability"
_ICLR.cc/2019/Workshop/LLD — Submitted to LLD 2019_

### Official Review · AnonReviewer2 · 2019-04-06
**A nice paper studying angles of vectors through layers for data separation**

**Rating:** 4
**Confidence:** 2

**Review:**

The authors here present an interesting analysis on how skip-connections in residual networks and batch-normalization affect data separation. Their analysis included observing the transformation of the input vectors through hidden layers of the neural networks, like a standard multilayer perceptron, a resnet and a resnet with batch normalization. They did that by studying the angle and cosine similarity through the layers. This property is critical to decide if the model is able to separate points in different classes.

The paper is well written and easy to read. The settings and configurations are carefully explained and carry the detail needed to understand the analysis. The study of angles and cosine similarities between the layers is very interesting, although its relation to the data separation property (section 3.3) could be written in a more clear way.

Pros:
- connecting the dynamics of angles with data separation
- extensive analysis

Cons:
- more settings could be explored

Some questions that the authors could address as well: why are the specific resnet models selected by the authors (Yang, Hardt)? what are the effects of using different initialization methods for the internal weights? how much important is the specific initialization chosen by the authors? what happens when the number of training examples increases or decreases?

---

### Official Review · AnonReviewer1 · 2019-04-11
**Angle preservation analysis in BN and ResNet shortcuts with questionable conclusion**

**Rating:** 2
**Confidence:** 2

**Review:**

Summary:

The authors argue, that BN and ResNet shortcuts encourage angle preservation throughout the layers.

Novelty:

The analysis of angle preservation seems novel to me.

Rating:

I do not think the paper makes a compelling point for why preservation of angle is a desirable property. Conservation of norm could be just as important and even that might not be important if classification would e.g. be based on the norm of a feature vector. I don't see why angles are a particularly desirable feature to preserve and the paper fails to make a strong point for this either. Thus, I vote for reject.

---

### Decision · Program_Chairs · 2019-04-16
**Acceptance Decision**

**Decision:**

Reject

**Comment:**

R1 had several issues with the arguments in the paper. The paper is also not a great fit for the workshop topic